# Enhancing Physicochemical Properties and Single Cell Performance of Sulfonated Poly(arylene ether) (SPAE) Membrane by Incorporation of Phosphotungstic Acid and Graphene Oxide: A Potential Electrolyte for Proton Exchange Membrane Fuel Cells

**DOI:** 10.3390/polym13142364

**Published:** 2021-07-19

**Authors:** Sung Kwan Ryu, Ae Rhan Kim, Mohanraj Vinothkannan, Kyu Ha Lee, Ji Young Chu, Dong Jin Yoo

**Affiliations:** 1Department of Energy Storage/Conversion Engineering of Graduate School (BK21 FOUR), Hydrogen and Fuel Cell Research Center, Jeonbuk National University, Jeonju 54896, Jeollabuk-do, Korea; hellomong35@naver.com; 2Department of Life Science, Jeonbuk National University, Jeonju 54896, Jeollabuk-do, Korea; carumiss@naver.com (K.H.L.); ebbuneg@hanmail.net (J.Y.C.); 3R&D Education Center for Whole Life Cycle R&D of Fuel Cell Systems, Jeonbuk National University, Jeonju 54896, Jeollabuk-do, Korea; vinothkannanram@gmail.com

**Keywords:** hydrocarbon membrane, inorganic nanofiller, proton conductivity, ion cluster, PEMFC

## Abstract

The development of potential and novel proton exchange membranes (PEMs) is imperative for the further commercialization of PEM fuel cells (PEMFCs). In this work, phosphotungstic acid (PWA) and graphene oxide (GO) were integrated into sulfonated poly(arylene ether) (SPAE) through a solution casting approach to create a potential composite membrane for PEMFC applications. Thermal stability of membranes was observed using thermogravimetric analysis (TGA), and the SPAE/GO/PWA membranes exhibited high thermal stability compared to pristine SPAE membranes, owing to the interaction between SPAEK, GO, and PWA. By using a scanning electron microscope (SEM) and atomic force microscope (AFM), we observed that GO and PWA were evenly distributed throughout the SPAE matrix. The SPAE/GO/PWA composite membrane comprising 0.7 wt% GO and 36 wt% PWA exhibited a maximum proton conductivity of 186.3 mS cm^−1^ at 90 °C under 100% relative humidity (RH). As a result, SPAE/GO/PWA composite membrane exhibited 193.3 mW cm^−2^ of the maximum power density at 70 °C under 100% RH in PEMFCs.

## 1. Introduction

Renewable energy devices have received a great deal of attention due to the depletion of fossil fuels and environmental pollution. Among the renewable energy devices being explored, polymer electrolyte membrane fuel cells (PEMFCs) have been studied extensively for stationary and portable applications due to their high power output, low noise, and zero emission of environmental pollutants [1,2,3,4]. In particular, PEMFCs produce highly efficient and environmentally friendly energy because they use hydrogen and oxygen as fuels to produce electricity through a chemical reaction and generate water as a by-product [5,6,7,8]. The polymer electrolyte membrane (PEM) of a PEMFC transfers protons between the electrodes and prevents mixing of H_2_ and O_2_ gases, and is, therefore, the primary component of PEMFCs. DuPont Nafion^®^ is a widely used, commercially available PEM that contains sulfonic acid (–SO_3_H) groups and shows good performance under humidifying conditions, high proton conductivity, and good thermomechanical properties [9,10]. However, perfluorosulfonic acid membrane (PFSA) based membranes are very expensive to manufacture and have problems such as CO poisoning and operating temperature limits due to a low T_g_. This has led many scientists to research alternative membranes with higher performance than Nafion, including hydrocarbon membranes [11,12,13]. Hydrocarbon membranes have many advantages such as a simple structure, thermal and mechanical stability, low gas permeability, high water content, various operating temperatures due to a high T_g_, and low manufacturing cost [14,15,16]. However, hydrocarbon membranes have lower proton conductivity than PFSA membranes [17,18,19,20]. To address this issue, researchers have investigated hydrocarbon polymers with a high degree of sulfonation (DS) [21,22]. However, when the DS of a polymer increases, it becomes difficult to use in a PEMFC system because the mechanical strength of the membrane decreases under the operating conditions of a fuel cell due to swelling. Thus, to overcome the limitations of a membrane with a high DS, TiO_2_, SiO_2_, Fe_2_TiO_5_, ZrO_2_, and graphene oxide (GO) have been added to the polymer matrix [23,24,25,26,27]. Kumar et al. fabricated a composite membrane by blending poly(vinylidene fluoride-*co*-hexafluoropropylene) and sulfonated TiO_2_ with different degrees of sulfonation. They reported that a 25% sulfonated TiO_2_ composite membrane showed the best performance, with a proton conductivity of 3.6 × 10^−3^ S cm^−1^. In addition, they reported that the sulfonated TiO_2_ used in the fabrication of the composite membrane formed hydrophilic ion channels thereby improving the conductivity of the membrane [28]. Lee et al. developed composite membranes comprising sulfonated poly(arylene ether ketone) and functionalized SiO_2_ and analyzed their properties. Field emission scanning electron microscopes (FE-SEM) analysis revealed uniform dispersal of functionalized SiO_2_ on the sulfonated poly(arylene ether ketone) (SPAEK) surface due to hydrogen bonding between –SO_3_H and functionalized SiO_2_ of the polymer, and the proton conductivity of the SPAEK composite film with functionalized SiO_2_ was 199.8 S cm^−1^ at 90 °C and 100% RH [29]. Yoo et al. reported improvement in the thermal and mechanical stability of composite membranes when GO was used as an inorganic nano-filler. In addition, the use of functionalized GO as a filler in the composite membrane resulted in excellent mechanical stability, fuel crossover prevention of membrane swelling, and improved ionic conductivity through π–π interactions with the polymer backbone [30].

Heteropoly acid (HPA) is an inorganic material generally suitable for use in PEMs due to its high ionic conductivity and thermal stability in amorphous form. In particular, HPA forms ion clusters and improves proton conductivity by binding with water molecules in the membrane [31,32,33,34]. Lu et al. produced a multilayer membrane impregnated with PWA using a mesoporous Nafion membrane (meso-Nafion). Nafion 115 membrane had a proton conductivity of 0.015 S cm^−1^ under low humidity (40% RH) conditions, while a multilayer membrane (meso-Nafion) had a proton conductivity of 0.072 S cm^−1^, representing a four-fold improvement in conductivity. These authors reported that leaching of PWA did not occur due to the structure of the new multilayer membrane [35]. HPA can improve membrane performance even under low-humidity and high-temperature experimental conditions. Zhang et al. prepared a polyethersulfone-polyvinylpyrrolidone (PES-PVP) composite membrane impregnated with phosphoric acid (PA) for application in HT-PEMFCs, and incorporated phosphotungstic acid (PWA) in the membrane to create an organic–inorganic composite membrane. PWA was uniformly dispersed on the surface of PA-(PES-PVP) composite membrane, and Fourier transform infrared spectrometer (FT-IR) analysis revealed strong interactions between PA and PWA. In addition, when 5 wt% PWA was added to the PA-(PES-PVP) composite membrane, a proton conductivity of 1.44 × 10^−1^ S cm^−1^ and power density of 416 mW/cm^2^ at 160 °C with no humidification were achieved [36]. Wang et al. produced composite membranes comprising sulfonated poly(ether ether ketone sulfone) (SPEEK) and HPA. They controlled the sulfonation degree of SPEEK and varied the content of HPA, and assessed the performance and properties of the resulting membranes at high temperatures (<100 °C). The degree of sulfonation of the SPEEKS/HPA composite membranes was 0.8 due to the strong interaction between sulfonic acid groups and HPA particles, and the HPA particle size was 10–30 nm. In addition, the thermal stability of the composite membranes was increased by the addition of HPA, and the proton conductivity was 0.068 S cm^−1^ at room temperature (25 °C) and 0.095 S cm^−1^ at 120 °C [37]. The above studies indicate that PWA can improve the performance of fuel cell composite membranes under high temperatures in the absence of humidification. In this study, we fabricated sulfonated poly(arylene ether) (SPAE) containing sulfonated fluorenyl units. To improve the mechanical stability of the synthesized polymer, we added GO to the membranes. In addition, to increase the proton conductivity of the SPAE membranes, we prepared composite membranes by solution blending with different amounts of PWA. The chemical structure and properties of the SPAE/GO/PWA composite membranes were observed using FT-IR, proton nuclear magnetic resonance (^1^H NMR), and thermogravimetric analysis (TGA), and the surface morphology and crystallinity of the membrane were examined through atomic force microscopy (AFM), FE-SEM, and X-ray diffraction (XRD) analyses. Proton conductivity and water uptake of the SPAE-PWA composite membranes were measured at temperatures ranging from 30 to 90 °C and 100% humidification in addition to the ion exchange capacities (IECs) of the SPAE composite membranes. Finally, in order to find out the suitability of prepared membranes for fuel cell applications, the PEMFC test was conducted at 70 °C under 100% RH.

## 2. Experimental

### 2.1. Materials

4,4′-(Hexafluoroisopropylidene)diphenol (BPHF, >97%), 4,4′-(9-fluorenylidene)-diphenol (BPFL, >97%), decafluorobiphenyl (DFBP, >99%), anhydrous potassium carbonate (K_2_CO_3_), anhydrous *N*,*N*-dimethylacetamide (DMAc, >99%), sulfuric acid (>98%), anhydrous toluene (>99%), chlorosulfonic acid (>99%), dichloromethane (>98%), and phosphotungstic acid hydrate (>99%) were purchased from the Sigma-Aldrich (Merck Korea), Gangnam-Gu 06178, Seoul, Republic of Korea and were used without further purification.

### 2.2. Synthesis of Polymers

#### 2.2.1. Synthesis of Poly(arylene ether) Block Copolymers

Perfluorinated poly(arylene ether) copolymer was synthesized by nucleophilic aromatic substitution reaction of BPFL, DFBP, and 6F-BPA in DMAc solvent. As shown in Scheme 1, BPFL (0.50 g, 1.497 mmol), DFBP (1.00 g, 2.994 mmol), 6F-BPA (0.52 g, 1.497 mmol), K_2_CO_3_ (0.45 g, 3.293 mmol), and DMAc (17.0 mL) were added to a 100 mL round-bottom (RB) flask, and the mixture was stirred under a nitrogen atmosphere. Then, the reaction temperature was increased to 95 °C under continuous stirring, and the temperature was maintained for 24 h. The solution was then poured into a large excess of solvent (methanol/DI water: 6/1) to precipitate the polymer powder, which was subsequently filtered. Afterward, the obtained powder was washed several times with DI water and methanol, and then dried in a vacuum oven at 60 °C for 24 h.

#### 2.2.2. Sulfonation of Poly(arylene ether) Block Copolymers

First, 2 g of polymer was dissolved in 20 mL of dichloromethane in a round flask. Afterward, 3 mL of chlorosulfonic acid and 20 mL of dichloromethane were slowly added to the round flask and reacted at room temperature for 24 h. After the reaction was completed, the reactant was added to cold distilled water to cool it, followed by washing and filtering with deionized (DI) water, and the collected polymer was dried at 80 °C for 24 h.

#### 2.2.3. Fabrication of SPAE/GO/PWA Composite Membranes

0.4 g of SPAE polymer, 0.7 wt% of GO, and 1 to 36 wt% PWA were dispersed in DMAc. After sufficient stirring using an ultrasonication device for 24 h to disperse the GO, the polymer solution was cast onto a flat glass plate to prepare thin film-shaped membranes. After drying the polymer solution in a vacuum oven at 110 °C for 24 h, 50–60 μm brown transparent composite membranes were collected. Figure 1 shows the structure of the fabricated composite membranes.

### 2.3. Chraterizations

Polymer functional groups were confirmed using Fourier transform infrared spectroscopy (FT-IR, PerkinElmer, Spotlight 4000, Waltham, MA, USA) analysis. Measurements were performed in the range of 500–4000 cm^−1^ with KBr pellets. To analyze the chemical structure of the composite membranes, ^1^H NMR (JEOL Ltd., JNM-ECA600, Tokyo, Japan) was performed, and samples were prepared using DMSO-d6. In addition, to determine the crystallinity of the composite films, X-ray diffractometry (XRD, Rigaku, D-Max-3A, Tokyo, Japan) was performed in the range of 5 to 90 °C. The thermal stability of composite membranes was evaluated using a thermal gravimetric analysis system (TGA, TA Instruments, Q50, New Castle, DE, USA), and measurements were taken by heating samples at 10 degrees per minute in the temperature range from 30 to 800 °C under a nitrogen atmosphere. Field emission scanning electron microscopy (FE-SEM, SUPRA 40VP) and energy dispersive X-rays spectroscopy (EDS) were used to observe the surface morphology of the composite films and assess the dispersibility of GO. In addition, the surface shape and roughness of the composite membranes were examined using atomic force microscopy (AFM, Seiko Instrument Co., SPA-300HV, Chiba, Japan).

Proton conductivity of SPAE/GO/PWA composite membranes was measured using a Bekk-Tech conductivity measurement cell combined with a PGZ301 EIS voltammetry device. To measure conductivity, 3.0 mm × 0.5 mm composite membrane samples were soaked in DI water for 48 h, and proton conductivity was then measured from 10 Hz to 100 kHz. Proton conductivity (σ) was calculated using the following equation:σ/S cm = L/(R × T × W)(1)
where σ is proton conductivity, L (cm) is the distance between the two electrodes, R (Ω) is the membrane resistance, T (µm) is the membrane thickness, and W (cm) is the width of the fully hydrated membrane.

The activation energy (E_a_) of composite films was calculated using the following equation, where R and T denote the gas constant and Kelvin temperature, respectively:lnσ = lnσ_0_ − E_a_ / (R × T)(2)

The ion exchange capacity (IEC, mequiv g^−1^) of composite membranes was measured using a titration method. To exchange H^+^ with Na^+^, 0.1 g of each composite membrane was immersed in 40 mL of 2 mol/L NaCl aqueous solution for 48 h at room temperature. Each sample was titrated with 0.01 mol NaOH aqueous solution until the pH reached 7 using phenolphthalein as an indicator. IEC was calculated using the following equation:IEC (mequiv g^−1^) = (V_NaOH_ − C_NaOH_)/W_dry_(3)
where V_NaOH_ is the volume of consumed NaOH (mL), C_NaOH_ is the concentration of NaOH, and W_Dry_ is the weight of the dry membrane.

After drying SPAE/GO/PWA composite membranes in a vacuum oven for 24 h, weight was measured, and composite membranes were placed in DI water for 24 h. After removing moisture from the surfaces of the composite membranes, weight was remeasured. Water uptake was calculated using the following equation:Water uptake (%) = (W_wet_ − W_dry_)/W_dry_ × 100% (4)
where W_Wet_ is the weight of the wet membrane and W_Dry_ is the weight of the dry membrane.

The swelling ratio of composite membranes was calculated by measuring changes in the height and length of membrane samples before and after water absorption for 24 h. The swelling ratio of the membranes was calculated using the following equation:Swelling ratio (%) = (S_wet_ − S_dry_)/S_dry_ × 100%(5)
where H_wet_ is the height of the wetted composite membrane and H_dry_ is the height of the dried composite membrane.

The contact angle was measured by analyzing the surface wettability of membranes. Depending on the hydrophilic or hydrophobic properties of the membrane, the contact angle also changes. After drying the composite membrane at 90 °C for 24 h, the degree of hydrophilicity of membrane samples was investigated using a contact angle measuring device (Smart Drop Plus). The mechanical properties of the composite membranes with a fully hydrated state, in particular the tensile strength and elongation at break, were evaluated using a universal testing machine (UTM, LR5K-plus) at a cross head speed of 10 mm min^−1^ at room temperature. The sample was cut into a dumbbell shape for the experiment.

### 2.4. Preparation of Membrane Electrode Assembly and Measurement of Single-Cell Performance

Membrane electrode assemblies (MEA) with an area of 5 cm^2^ were manufactured using commercial gas diffusion electrodes (GDEs, Pt/C, 0.3 mg/cm^2^, NARA CELL-TECH). SPAE/GO/PWA composite membrane was placed between two GDEs, and individual MEAs were fabricated using a hot-pressing method at 120 °C and 1800 psi for 3 min. Single cells were manufactured using components such as an end plate, separator plate, gasket, and MEA, and unit cells were manufactured at 70 kgf/cm^2^ pressure. The performance of unit cells was measured from open circuit voltage (OCV) to 0.3 V using a fuel cell performance evaluation device (Bekk-Tech). One hundred percent humidified hydrogen and air were injected into the unit cell based on a stoichiometry of 1.2:2.0, respectively, and the performance of unit cells was measured at 70 °C and ambient pressure.

## 3. Results and Discussion

### 3.1. Structural Characterization

SPAE membranes were successfully synthesized through polycondensation and sulfonation reactions [38]. The chemical structure of SPAE membranes was determined by ^1^H NMR and FT-IR analyses; the results of ^1^H NMR analyses are shown in Figure 2. Proton peaks assigned to the PAEK polymer chain were found between 6.8 and 8.2 ppm. These chemical shifts are in good agreement with those reported previously [39].

Figure 3 shows the FT-IR spectra measured at room temperature for PAE, SPAE, and SPAE/GO/PWA membranes. The 1000–1500 cm^−1^ absorption band displayed in all samples indicates a C–F stretching peak, along with C–C stretchin vibrations of the aromatic backbone. In addition, the two peaks 1023 and 1070 cm^−1^ found only in the SPAE sample are absorption bands due to SO_3_H generated through sulfonation. The stretching peak of 1000–1480 cm^−1^ reflects S=O bonding, while that at 1000–1480 cm^−1^ reflects S=O bonds. The 1480–1500 cm^−1^ peak was attributed to the C=C bond of aromatic groups, and the peak at 3400 cm^−1^ to stretching vibrations of OH bonds. The FT-IR results of SPAE and SPAE/GO/PWA composite membranes were similar, indicating that no chemical bonds formed between SPAE and PWA [40,41].

### 3.2. Thermal Stability

Before TGA measurements, samples were heat-treated at 100 °C for 12 h in a vacuum oven to remove free water and residual solvent from the membranes. Thermal stability of the membranes was analyzed using TGA in a nitrogen atmosphere at a heating rate of 10 °C min^−1^, and the results are shown in Figure 4. Three-stage weight loss was observed [42]. The temperature at which decomposition occurred shifted slightly depending on the PWA content of the composite membrane, but no significant differences in the thermal stability of composite membranes were observed. The first weight loss of SPAE/GO/PWA composite membranes occurred in the range of approximately 100 to 200 °C, and we attribute this to loss of water adsorbed on the sulfonic acid groups and PWA of the composite membranes. Further weight loss was observed at about 300 °C due to loss of the sulfonic acid groups of the composite membranes, and there was additional weight loss at 500 °C due to decomposition of the polymer backbone. These TGA results indicate that the SPAE/GO/PWA composite membranes were thermally stable up to 250 °C.

### 3.3. XRD Analysis

The microstructures of GO, PWA, and SPAE/GO/PWA composite membranes were assessed using XRD analysis, as shown in Figure 5. Several sharp peaks were observed for PWA powder as shown in Figure 5a, indicating that the PWA was highly crystalline. In addition, the XRD pattern of graphene oxide showed a characteristic crystalline diffraction peak around 2θ = 10° [43,44]. Similar XRD measurement results were obtained for pristine SPAE membrane and SPAE/GO/PWA composite membrane Figure 5b. Both pristine SPAE membrane and SPAE/GO/PWA composite membrane had a wide diffraction peak of approximately 2θ between 15–25°, indicating an amorphous structure. The XRD pattern of the SPAE/GO/PWA composite membrane showed a crystalline diffraction peak of GO around 2θ = 10°, indicating that GO was properly dispersed in the composite membrane. As the PWA content of the SPAE/GO/PWA composite membranes increased, the intensity and the crystallinity of the composite membranes shifted slightly to the right Figure 5. This indicates that the inorganic nanofillers were homogenously dispersed due to the amorphous nature of the composite membranes.

### 3.4. Morphologies of the Composite Membranes

The morphologies of SPAE/GO/PWA composite membranes with inorganic fillers such as GO and PWA were determined by FE-SEM and EDS analyses, and the results are shown in Figure 6. The corresponding EDS mapping images are shown in Appendix A. The surface of the pure SPAE membrane was flat and without any defects, whereas the surface of SPAE/GO/PWA membranes was rough due to the limited dispersion of GO. When inorganic materials are introduced into a polymer matrix, the surface shape and pores of the membrane may be deformed due to interfacial interactions. PWA should disperse evenly in a polymer matrix due to the physical action of some of the hydrophilic groups of the SPAE polymer. The presence of PWA in the composite membrane was observed through EDX analysis Figure 6. Pristine SPAE membrane was composed of carbon, oxygen, fluorine, and sulfur while SPAE/GO/PWA composite membranes were composed of carbon, oxygen, fluorine, sulfur, and tungsten. The EDX analysis results indicated that PWA particles were completely dispersed in the matrix of the SPAE membrane. Two- and three-dimensional images of pristine membrane and SPAE composite membranes were obtained by AFM and results are shown in Figure 7. The microstructure of SPAE composite membranes was clearly separated into a bright region corresponding to hydrophobic polymer backbones and dark regions related to hydrophilic SO_3_H groups and PWA. In the AFM image of the pristine SPAE membrane shown in Figure 7a, partially densely formed ion channels and clusters were observed, indicating poor connections between ion channels. However, AFM images of SPAE/GO/PWA composite membranes revealed that as the content of PWA increased, hydrophilic and hydrophobic areas became more closely connected, indicating that ion channels were evenly distributed on the surface of the membranes. This type of distribution of ion clusters increases proton conductivity.

### 3.5. Water Uptake, Swelling Ratio, and Ion Exchange Capacity (IEC)

In PEMFCs, water plays an important role in smoothly moving protons from the anode to the cathode, and most of the water uptake by a membrane is related to IEC. High water uptake can improve the proton conductivity of PEMFCs, but if the membrane absorbs too much water, the mechanical strength of the membrane is greatly reduced, making it difficult for the membrane to retain its shape [45]. Figure 8a shows the water uptake of membranes at different temperatures according to PWA content. In SPAE/GO/PWA composite membranes, as the PWA content increased, water uptake gradually increased, because the content of hydrophilic functional groups increased. PWA in the composite membranes helped improve ion clusters and interactions between the adsorbed water and matrix of the composite membrane. Furthermore, the swelling ratio of the composite membranes showed a tendency similar to water uptake according to temperature as shown in Figure 8b. When PWA was added to the SPAE membrane up to 36 wt%, the swelling ratio of the membrane increased from 8% to 22% at 90 °C, which indicates excellent dimensional stability of the SPAE/GO/PWA composite membranes. Ion exchange capacity (IEC) is an important indicator of the content of sulfonic acid groups in a polymer electrolyte membrane. As the DS increases, the IEC value of membranes increases. Figure 9c shows IEC values according to PWA content of the composite membranes. As the PWA content of the composite membranes increased, the IEC value increased from 1.68 mequiv g^−1^ to 2.02 mequiv g^−1^, and PWA played an important role in increasing proton exchange in the composite membranes.

### 3.6. Water Contact Angle and Mechanical Properties

The water contact angle is the angle between the membrane and water droplets dropped on the membrane. From Figure 9, it can be observed that the water contact angle decreased from 72.67° to 61.55° as the PWA content of the membrane increased. This means that PWA in SPAE membranes combines with moisture to reduce surface strength and increase the moisture absorption capacity of the composite membrane [46]. In other words, the water contact angle is closely related to IEC, water uptake, and proton conductivity.

The mechanical properties of fully hydrated SPAE/GO/PWA composite membranes provide an indication of membrane durability. In general, fully hydrated membranes absorb water molecules resulting in dimensional changes that can adversely affect the mechanical properties of the membrane. To compare the physical properties of membranes according to PWA content, the mechanical properties of composite membranes were evaluated in the fully hydrated state, and the results are shown in Table 1. Among the mechanical properties of prepared SPAE/GO/PWA composite membranes, tensile strength increased from 13.2 to 20.4 MPa as the content of PWA increased. In addition, Appendix A shows the stress–strain graphs of SPACE composite membranes in the Appendix A. We attributed this to interactions (π–π interactions) between PWA and the polymer backbone, resulting in excellent mechanical stability of the SPAE/GO/PWA composite membranes.

### 3.7. Proton Conductivity and Activation Energy

Proton conductivity is a critical determinant of the performance of PEMFCs, and is generally influenced by IEC and water uptake. The proton conductivity of the prepared SPAE/GO/PWA composite membranes was measured in the temperature range from 30 to 90 °C at 100% humidification using an AC impedance device with a four-electrode system. Membrane samples were dried in an oven at 80 °C for 24 h, and then the thickness of each membrane was measured and free acid was removed by immersing the membranes in distilled water for 12 h before proton conductivity measurements. Figure 10 shows the proton conductivity of SPAE/GO/PWA composite membranes according to temperature at 100% RH. Proton conductivity of the SPAE/GO/PWA composite membranes increased with an increase in temperature. This is because the higher the temperature, the faster the diffusion of water in the SPAE membrane and the higher the moisture absorption capacity, resulting in enhanced proton conductivity of the polymer membrane itself. In addition, PWA played a very important role in increasing the proton conductivity of the membrane. Compared to pristine SPAE, the SPAE/GO/PWA composite membranes exhibited higher proton conductivity due to the formation of ionic clusters between PWA and water molecules and the hydrophilic matrix of the polymer. Appendix A shows the characteristics of the SPAE composite membrane in the Appendix A. The conductivity of SPAE/GO/PWA composite membrane improved as the content of PWA increased. Pristine SPAE membrane had a proton conductivity of 105 mS cm^−1^ at 90 °C, while SPAE-PWA (36 wt%) membrane had a conductivity of 186 mS cm^−1^ at 90 °C.

The activation energy (E_a_) of proton transfer can be used to evaluate the mechanism of proton transfer [47,48]. In general, if the E_a_ value is less than 14, vehicle-type proton transport is occurring, while if E_a_ is between 14 and 40, Grotthuss-type proton transfer is occurring. As shown in Figure 11, the E_a_ value of the SPAE/GO/PWA composite membrane ranged from 8.47 to 11.34 kJ/mol, and as the PWA content increased, E_a_ gradually decreased, indicating that PWA helped the movement of ions. In addition, the E_a_ value of all SPAE/GO/PWA composite membranes was consistent with a vehicle mechanism of proton transport, indicating that the introduction of PWA into the SPAE matrix resulted in more facile movement of protons in the membrane.

### 3.8. Unit Cell Performance

After fabricating MEAs containing pristine SPAE, SPAE/GO, SPAE/GO/PWA 24 wt%, and SPAE/GO/PWA (36 wt%) membranes, single cell performance was measured, and results are shown in Figure 12. To measure the performance of unit cells, a fuel cell performance evaluation device (Bekk-Tech) was used, and 100% humidified hydrogen and air were injected into the unit cell at a stoichiometry of 1.25:2.0 and 70 degrees, respectively. The OCVs of MEAs of pristine SPAE, SPAE/GO, SPAE/GO/PWA (24 wt%), and SPAE/GO/PWA (36 wt%) were 0.93 V, 0.94 V, 0.96, and 0.95 V, respectively. These results indicate that the SPAE MEAs had excellent hydrogen permeation resistance. The current densities of pristine SPAE, SPAE/GO, SPAE/GO/PWA (24 wt%), and SPAE/GO/PWA (36 wt%) measured at 0.6 V were 80 mA cm^−2^, 90 mA cm^−2^, 196 mA cm^−2^, and 276 mA cm^−2^, respectively, and the maximum power densities were 54 mW cm^−2^, 68 mW cm^−2^, 163 mW cm^−2^, and 193 mW cm^−2^, respectively. The performance of single cells containing SPAE/GO/PWA composite membrane was higher than that of pristine SPAE. Together, these results indicate that introduction of PWA into SPAE membranes results in smooth proton transport due to ion channels in the polymer matrix [49,50].

## 4. Conclusions and Discussion

Sulfonated poly(arylene ether) (SPAE) composite membrane containing sulfonated fluorenyl units was prepared by varying the content of GO and PWA. FT-IR, XRD, and TGA analyses indicated that PWA was well coupled to the polymer matrix, and FT-IR and TGA analyses revealed that composite membranes had the expected chemical structures and good thermal stability up to 250 °C. FE-SEM images revealed that GO and PWA was evenly distributed on the membrane surface. As the PWA content in the composite membranes increased, water uptake increased to 52%, whereas no significant dimensional changes of SPAE/GO/PWA composite membranes were observed. All composite membranes fabricated using PWA had a high conductivity of 113–186 mS cm^−1^ (IEC 1.72–2.02 mequiv g^−^^1^) under 100% RH and showed significantly improved proton conductivity compared to the pristine SPAE membrane due to hydrogen bonding between PWA and the polymer matrix. As a result of a single cell test of composite membranes containing PWA, the maximum current density and power density was 276 mA cm^−2^ (at 0.6 V) and 193.3 mW cm^−2^, and the performance was significantly improved compared to the pristine SPAE membrane. Therefore, the incorporation of inorganic materials into the polymer matrix can improve the performance of PEMFCs.

## Data Availability

Not applicable.

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
