# Peer review of "Enhancing Physicochemical Properties and Single Cell Performance of Sulfonated Poly(arylene ether) (SPAE) Membrane by Incorporation of Phosphotungstic Acid and Graphene Oxide: A Potential Electrolyte for Proton Exchange Membrane Fuel Cells"

_polymers, 2021, doi:10.3390/polym13142364_

Round 1

Reviewer 1 Report

The reviewed manuscript investigates the physicochemical enhancement ability of adding phosphotungstic acid (PWA) and graphene oxide (GO) for sulfonated poly (arylene ether) (SPAE) membrane on the fuel cell performance. The article is made at a good scientific and technical level, and its practical significance is beyond doubt. In order to improve the readability and clarity of the manuscript, some major concerns need to be addressed before the paper is to be accepted for publishing:

  1. The manuscript title is little confused, especially “as filler have”, please try to reconstruct it in a clearer sentence.
  2. The abstract is a mini version of manuscript that proceeds. So, include introduction, methodology, results & Discussion and concluding remarks in a precise brief but effective manner.
  3. Please identify all peaks in XRD pattern, Figure 5.
  4. Figure 5 (XRD), please provide the unique powder diffraction file (PDF) for each element/compound in the pattern.
  5. Figure 6 (SEM), scale bar is not clear. Please redraw.
  6. Line 279: “When inorganic materials are introduced into a polymer matrix, the surface shape and pores of the membrane may be deformed due to interfacial interactions”, better to support this claim by high magnification SEM image, at least for the highest concentration (36 wt%), around 10k magnification or above.
  7. Line 285: “The EDX analysis results indicated that PWA particles were completely dispersed in the matrix of the SPAE membrane.” Actually, EDX can’t confirm the homogeneous distribution in the matrix unless you test many spots and take the average results. Better to use mapping to proof that homogeneity.
  8. Figure 6 (EDX), please include the elements percentages on each image. Also, the tested or scanned area/spot/point should be indicated.
  9. Line 350: What was the parameters and conditions of the tensile strength? Where are the machine specifications? According which standard the test was performed? Please include the stress-strain curves in one graph.
  10. Please provide a graphical abstract if it is possible.

Author Response

Reviewer 1#

Comments: The reviewed manuscript investigates the physicochemical enhancement ability of adding phosphotungstic acid (PWA) and graphene oxide (GO) for sulfonated poly (arylene ether) (SPAE) membrane on the fuel cell performance. The article is made at a good scientific and technical level, and its practical significance is beyond doubt. In order to improve the readability and clarity of the manuscript, some major concerns need to be addressed before the paper is to be accepted for publishing.

Response: Thanks to the reviewer for the kind encouragement and valuable suggestions to improve the manuscript. According to reviewer’s comments, we have revised the manuscript and all changes made in revised manuscript were indicated by blue colored letters.

Question 1: The manuscript title is little confused, especially “as filler have”, please try to reconstruct it in a clearer sentence.

Response: Thanks to the reviewer for valuable suggestion. As per reviewer suggestion, the title of the paper was clearly revised to avoid confusion of readers. Instead of ambiguous sentences such as "as filler have", the entire title of the paper was revised as shown below.

“Novel composite membrane containing phosphotungstic acid and graphene oxide as filler have enhanced physicochemical properties and membrane electrode assembly performance for fuel cell applications.” has been revised to “Enhancing physicochemical properties and single cell performance of sulfonated poly (arylene ether) (SPAE) membrane by incorporation of phosphotungstic acid and graphene oxide: A potential electrolyte for proton exchange membrane fuel cells.” (Please see the page no. 1 in revised manuscript)

Question 2: The abstract is a mini version of manuscript that proceeds. So, include introduction, methodology, results & discussion and concluding remarks in a precise brief but effective manner.

Response: Thanks to the reviewer for critical comment. Based on the reviewer comment, the abstract has been elaborated by including methodology of membrane preparation, results and concluding remarks as shown below.

“Development of novel proton exchange membrane (PEM) is imperative for further commercialization of proton exchange membrane fuel cells (PEMFCs). In this work, phosphotungstic acid (PWA) and graphene oxide (GO) were integrated into sulfonated poly (arylene ether) (SPAE) to create a composite membrane for PEMFCs. Incorporation of PWA enhanced membrane hydrophilicity, while GO improved the mechanical strength of the membrane. Both PWA and GO improved the structural, morphological, thermal, mechanical, oxidative, and electrochemical properties of SPAE composite membranes. Thus, SPAE/GO/PWA composite membrane exhibited improved proton conductivity and PEMFC power and current output compared with bare SPAE membrane.” has been revised to “Development of potential and novel proton exchange membranes (PEMs) is imperative for further commercialization of PEM fuel cells (PEMFCs). In this work, phosphotungstic acid (PWA) and graphene oxide (GO) were integrated into sulfonated poly (arylene ether) (SPAE) through solution casting approach to create a potential composite membrane for PEMFC applications. Thermal stability of membranes was observed using thermogravimetric analysis (TGA), and the SPAE/GO/PWA membranes exhibited high thermal stability compared to pristine SPAE membrane, owing to the interaction between SPAEK, GO and PWA. By using scanning electron microscope (SEM) and atomic force microscope (AFM), we observed that GO and PWA were evenly distributed throughout the SPAE matrix. The SPAE/GO/PWA composite membrane comprising 0.7 wt% GO and 36 wt% PWA exhibited a maximum proton conductivity of 186.3 mS cm-1 at 90 ℃ under 100 % relative humidity (RH). As a result, SPAE/GO/PWA composite membrane exhibited 193.3 mW cm-2 of the maximum power density at 70 ℃ under 100 % RH in PEMFCs”. (Please see the page no. 1 in revised manuscript)

Question 3: Please identify all peaks in XRD pattern, Figure 5.

Response: Thanks to the reviewer for important suggestion. According to the reviewer comment, all peaks were mentioned in XRD patterns (Figure 5) as shown below.

has been revised to

(Please see the page no. 8 in revised manuscript)

Question 4: Figure 5 (XRD), please provide the unique powder diffraction file (PDF) for each element/compound in the pattern

Response:  Thanks to the reviewer for the valuable comment. Actually, we have identified the individual peaks in GO and PWA using the following references. So, we ensured that the formation of successful elements.

  1. Sun, F.; Qin, L. L.; Zhou, J.; Wang, Y. K.; Rong, J. Q.; Chen,Y. J.; Ayaz, S.; Hai-Yin, Y. ; Liu, L. Friedel-crafts self-crosslinking of sulfonated poly(etheretherketone) composite proton exchange membrane doped with phosphotungstic acid and carbon-based nanomaterials for fuel cell applications: J. Membr. Sci. 2020, 611, 118381-118390, Doi.org/10.1016/j.memsci.2020.118381.
  2. Peng, Q.; Li, Y; Qiu, M.; Shi, B.; He, X.; Fan, C.; Mao, X.; Wu, H.; Jiang, Z Enhancing proton conductivity of sulfonated poly(ether ether ketone) based membranes by incorporating phosphotungstic acid coupled graphene oxide: Ind. Eng. Chem. Res. 2021, 60, 4460-4470, Doi.org/10.1021/acs.iecr.1c00003.
  3. Chen, P.; Li, H.; Song, S.; Weng, X.; He, D.; Zhao, Y. Adsorption of dodecylamine hydrochloride on graphene oxide in water: Results phys. 2017, 7, 2281-2288, Doi.org/10.1016/j.rinp.2017.06.054.

Question 5:. Figure 6 (SEM), scale bar is not clear. Please redraw.

Response: We agree with the reviewer suggestion. The scale bar is already marked on the FE-SEM images, but I think it is difficult to see because of the small size. As suggested by the reviewer, the scale bar was clearly marked while measuring the FE-SEM again. Also, we mentioned the scale bar in corresponding Figure caption.

Has been revised to

Figure 6. The SEM micrographs and EDX spectrum of SPAE composite membranes: a) SPAE prinstine membrane, b) SPAE/GO, c) SPAE/GO/PWA 12 wt%, d) SPAE/GO/PWA 36 wt%. Note that the scale bar for al l images is 1 µm. (Please see the page no. 9 in revised manuscript)

Question 6: Line 279: “When inorganic materials are introduced into a polymer matrix, the surface shape and pores of the membrane may be deformed due to interfacial interactions”, better to support this claim by high magnification SEM image, at least for the highest concentration (36 wt%), around 10 k magnification or above.

Response: We measured the FE-SEM again at 10K magnification as suggested by the review, and the EDS mapping result is also attached to the supporting information. Therefore, the even dispersion of PWA can be observed through the EDS results. (Please see the supporting information) 

Question 7: Line 285: “The EDX analysis results indicated that PWA particles were completely dispersed in the matrix of the SPAE membrane.” Actually, EDX can’t confirm the homogeneous distribution in the matrix unless you test many spots and take the average results. Better to use mapping to proof that homogeneity.

Response: Thanks to the reviewer for the valuable comment. Based on the reviewer comment, the EDS measurement were conducted for SPAE/GO/PWA membrane, and the obtained results were given in the Supporting Information.

Question 8: Figure 6 (EDX), please include the elements percentages on each image. Also, the tested or scanned area/spot/point should be indicated.

Response: In addition to the results of FE-SEM measured at high magnification, EDS results are shown in supporting information. element percentages can also observe.

Question 9: Line 350: What was the parameters and conditions of the tensile strength? Where are the machine specifications? According to which standard the test was performed? Please include the stress-strain curves in one graph.

Response: Thanks to reviewer for valuable comment. As per reviewer comment the measurement condition and curves of tensile strength was provided in revised Supporting Information as shown below.

“The mechanical properties of the composite membranes with fully hydrate state, in particular the tensile strength and elongation at break, were evaluated using a universal testing machine (UTM, LR5K-plus) at a cross head speed of 10 mm min-1 at room temperature. And test sample were cut into a dumbbell shape for the experiment.”

Figure S3. Stress-strain curves of SPAE composite membranes. (Please see the supporting information)

Question 10: Please provide a graphical abstract if it is possible.

Response: Thanks to the reviewer for critical comment. We produced a graphic abstract using the experimental results that can represent core of the manuscript.

Reviewer 2 Report

The authors develop of novel proton exchange membrane (PEM) for proton exchange membrane fuel cells (PEMFCs) by integrating phosphotungstic acid (PWA) and graphene oxide (GO) into sulfonated poly (arylene ether) (SPAE). A new SPAE composite membrane for PEMFCs is proposed to improve the structural, morphological, thermal, mechanical, oxidative, and electrochemical properties. The SPAE/GO/PWA composite membrane is tested and compared with bare SPAE membrane.
The conclusions are validated by the presented results . 

More than 50% of references are published in the last 5 years, highlighting the topicality of this research work.

Recommendations:
1) Avoid lumped references; a short comment should be included for each reference or two references in the same subject.

2) mention the objective and novelty of this work at the end of the introduction. 

3) structure of the paper should be presented at the end of the introduction.

4) include a discussion section, where the structural, morphological, thermal, mechanical, oxidative, and electrochemical properties of the SPAE/GO/PWA composite membrane are discussed and compared with bare SPAE membrane; use summarize tables of the results presented in the sections.

Author Response

Reviewer 2#

Comments: The authors develop of novel proton exchange membrane (PEM) for proton exchange membrane fuel cells (PEMFCs) by integrating phosphotungstic acid (PWA) and graphene oxide (GO) into sulfonated poly (arylene ether) (SPAE). A new SPAE composite membrane for PEMFCs is proposed to improve the structural, morphological, thermal, mechanical, oxidative, and electrochemical properties. The SPAE/GO/PWA composite membrane is tested and compared with bare SPAE membrane. The conclusions are validated by the presented results. More than 50% of references are published in the last 5 years, highlighting the topicality of this research work.

Response: Thanks to the reviewer for the kind encouragement and valuable suggestions to improve the manuscript. According to reviewer’s comments, we have revised the manuscript and all changes made in revised manuscript were indicated by blue colored letters.

Question 1: Avoid lumped references; a short comment should be included for each reference or two references in the same subject.

Response: Thanks to the reviewer for the valuable comment. Following the reviewer's kind advice, the reference was re-examined and corrected. Particularly, it was confirmed and revised that 50% of total references from recent 5 years.

Question 2: Mention the objective and novelty of this work at the end of the introduction.

Response: Thanks to the reviewer for critical comment. Based on the reviewer comment, the purpose and novelty of the work were included at the end of Introduction in revised manuscript.

Question 3: Structure of the paper should be presented at the end of the introduction.

Response: As in the answer to question 2, the manuscript was revised, and the purpose and structure of manuscript was written after the introduction.

“The above studies indicate that PWA can improve the performance of fuel cell composite membranes under high temperatures in the absence of humidification. In this study, we fabricated sulfonated poly (arylene ether) (SPAE) containing sulfonated fluorenyl units. To improve the mechanical stability of the synthesized polymer, we added GO to the membranes. In addition, to increase the proton conductivity of the SPAE membranes, we prepared composite membranes by solution blending with different amounts of PWA. The chemical structure and properties of the SPAE/GO/PWA composite membranes were observed using FT-IR, proton nuclear magnetic resonance (1H NMR) and thermogravimetric analysis (TGA), and the surface morphology and crystallinity of the membrane were examined through atomic force microscopy (AFM), FE-SEM, and X-ray diffraction (XRD) analyses. Proton conductivity and water uptake of the SPAE-PWA composite membranes were measured at temperatures ranging from 30 to 90 ℃ and 100% humidification in addition to the ion exchange capacities (IECs) of the SPAE composite membranes.” has been revised to “The above studies indicate that PWA can improve the performance of fuel cell composite membranes under high temperatures in the absence of humidification. In this study, we fabricated sulfonated poly (arylene ether) (SPAE) containing sulfonated fluorenyl units. To improve the mechanical stability of the synthesized polymer, we added GO to the membranes. In addition, to increase the proton conductivity of the SPAE membranes, we prepared composite membranes by solution blending with different amounts of PWA. The chemical structure and properties of the SPAE/GO/PWA composite membranes were observed using FT-IR, proton nuclear magnetic resonance (1H NMR) and thermogravimetric analysis (TGA), and the surface morphology and crystallinity of the membrane were examined through atomic force microscopy (AFM), FE-SEM, and X-ray diffraction (XRD) analyses. Proton conductivity and water uptake of the SPAE-PWA composite membranes were measured at temperatures ranging from 30 to 90 ℃ and 100% humidification in addition to the ion exchange capacities (IECs) of the SPAE composite membranes. Finally, in order to find out the suitability of prepared membranes for fuel cell applications, the PEMFC test was conducted at 60 ℃ under 100% RH. (Please see the page no. 3 in revised manuscript)

Question 4: Include a discussion section, where the structural, morphological, thermal, mechanical, oxidative, and electrochemical properties of the SPAE/GO/PWA composite membrane are discussed and compared with bare SPAE membrane; use summarize tables of the results presented in the sections.

Response: We revised the manuscript and summarized the experimental results in the conclusion section. In particular, we added the experimental results as table of proton conductivity, mechanical properties, and morphological properties in the supporting information.

Round 2

Reviewer 1 Report

The revision is satisfactory and the authors have provided amendments to all the suggested queries. Therefore, I recommend this work for publication in Polymers journal.